



# The Levantine Intermediate Water in the western Mediterranean and its interactions with the Algerian Gyres: insights from 60 years of observation

Katia Mallil[1,2], Pierre Testor[1], Anthony Bosse[3], Félix Margirier[4], Loic Houpert[5], Hervé Le Goff[1], Laurent Mortier[1], and Ferial Louanchi[2]

[1]Laboratoire d'Océanographie et du Climat: Expérimentations et Approches Numériques (LOCEAN, UMR 7159): CNRS/SU/MNHN/IRD, 75005, Paris, France
[2]Ecole Nationale Supérieure des Sciences de la Mer et de l'Aménagement du Littoral (ENSSMAL), Laboratoire des Ecosystèmes Marins et Littoraux (EcosysMarL), 16320, Alger, Algeria
[3]Aix-Marseille Université, Université de Toulon, CNRS, IRD, MIO UM110, 13288, Marseille, France
[4]School of Earth and Atmospheric Sciences, Georgia Institute of Technology, Atlanta, Georgia, USA
[5]OSE Engineering, 78470, Saint-Rémy-lès-Chevreuse, France

**Correspondence:** Katia Mallil (mallil.katia@gmail.com)

**Abstract.** The presence of two large scale cyclonic gyres in the Algerian basin influences the general and eddy circulation, but their effect on water mass transfer remain poorly characterized. Our study has confirmed the presence of these gyres using the first direct current measurements of the whole water column collected during the SOMBA-GE2014 cruise, specifically designed to investigate these gyres. Using cruise sections and a climatology from 60 years of *in situ* measurements, we have also shown the effect of these gyres on the distribution at intermediate depth of Levantine Intermediate Water (LIW) with warmer ($\sim$0.15° C) and saltier ($\sim$0.02 g.kg$^{-1}$) characteristics in the Algerian basin than in the Provençal basin. The Algerian gyres also impact horizontal density gradients with sinking of the isopycnals at the gyres' centres. Temporal cross-correlation of LIW potential temperature referenced to the signal observed south of Sardinia reveal timescale of transit of 4 months to get to the centre of the Algerian basin.

The LIW temperature and salinity trends over various periods are estimated to: +0.0017±0.0014° C.year$^{-1}$ and +0.0017 ±0.0003 year$^{-1}$ respectively over the 1960-2017 period, and accelerating to +0.059±0.072° C.year$^{-1}$ and +0.013 ±0.006 year$^{-1}$ over the 2013-2017 period.

## 1 Introduction

The Mediterranean Sea is a semi-enclosed evaporation basin with water and heat deficits (Béthoux, 1979; Bryden and Kinder, 1991). The dynamics of the Mediterranean Sea is characterized by an active thermohaline circulation resulting from a strong air/sea coupling and preconditionning to deep vertical mixing (Robinson et al., 2001). The difference in water density and sea level at Gibraltar Strait forces a surface inflow of warm and fresh Atlantic Waters (AW). Flowing then cyclonically along the continental slope of the different sub-basins (Millot, 1999; The MerMex Group: Durrieu de Madron et al., 2011). At the northern coast of Africa flows the Algerian Current, its high velocities up to 1 m/s (Benzohra and Millot, 1995) and important



baroclinic instabilities generates mesoscale meanders and Algerian Eddies (Millot et al., 1990). At depth, saltier and colder (on average over the year) waters exit through the same strait and form the Mediterranean Outflow Water (MOW) through cascading and mixing in the Atlantic. This exchange between the Atlantic Ocean and the Mediterranean Sea occurring in the southwestern Mediterranean is a major driver of the water messes dynamics of this basin but also the whole Mediterranean Sea (Robinson et al., 2001). The Levantine Intermediate Water (LIW) represents about half of the Mediterranean Outflow Water, the other half being made of Western Mediterranean Deep Water (WMDW) (Gascard and Richez, 1985; Bryden et al., 1994). MOW signature can be identified in the whole North Atlantic, and impacts the global thermohaline circulation (Johnson, 1997; Lozier and Stewart, 2008). Though, relatively few studies about ocean circulation have focused on the southwestern Mediterranean so far, compared to the northwestern Mediterranean where important ventilation and deep convection processes occur (MEDOC, 1970; Houpert et al., 2016; Testor et al., 2018). The Levantine Intermediate Water formed in the Eastern Mediterranean Sea lays between ∼300 and ∼700 m depth in the western basin and is identified by a temperature (and salinity) maximum. It appears in $\theta$-S diagrams as the so-called "scorpion tail" shape (Tchernia, 1958).

During the Mass Transfer and Ecosystem Response (MATER) program, isobaric floats drifting at 600m depth (from July 1997 to June 1998), moorings and profiling floats with nominal parking depths of 1200 and 2000m depth (1997 to 2002), were used to assess the circulation of LIW, Tyrrhenian Deep Water (TDW) and WMDW. The dominant pattern revealed by the float trajectories are two large-scale cyclonic gyres, so-called western and eastern Algerian Gyres, centred around [37°30' N, 2°3' E] and [38°30' N, 6° E], respectively (Testor et al., 2005b). These two gyres affect the whole water column and their location is strongly related to the closed $f/H$ contours ($f$ is the planetary vorticity and $H$, the water depth). The barotropic orbital velocities of the gyres are about 5 cm/s.

It has been shown through the ELISA (Eddies and Leddies Interdisciplinary Study in the Algerian Basin) experiment that Algerian Eddies (AEs) transport LIW from the vein flowing northward along the continental slope of Sardinia toward the interior of the basin by entrapping or dispatching pieces of it (Taupier-Letage et al., 2003; Millot and Taupier-Letage, 2005a). The LIW vein south of Sardinia can become unstable and generate anticyclonic eddies that can also transport LIW toward the interior of the basin (Millot, 1999; Testor and Gascard, 2005). The possibility of the presence of a permanent westward LIW vein across the Algerian basin as described by Wüst (1961) has been largely rejected by the scientific community in favour of an eddy-transport. Puillat et al. (2002) have used satellite images, mainly from NOAA/AVHRR thermal infrared channels (February 1996 to December 1998) to track Algerian Eddies over long term, and have evidenced their cyclonic circuit in the eastern part of the Algerian basin that could help in transporting LIW westward. This was further documented by Escudier et al. (2016a) showing a cyclonic mean path around both Algerian Gyres of the anticyclonic eddies tracked with their surface signature during 20 years (1993-2014) using satellite altimetry. Testor and Gascard (2005) have also observed the formation of Sardinia Eddies (SEs) transporting LIW in their cores in the centre of the basin and linked it to the detachment of the LIW vein further at the southwestern tip of Sardinia and the presence of a large scale cyclonic motion in the Algerian Basin (the Eastern Algerian Gyre). Testor et al. (2005a) have furthe investigated the formation of these eddies and assessed their impact on eddy transport using numerical modelling. In addition to AEs and SEs, Bosse et al. (2015) have shown the important contribution



to the spreading of LIW in the western Mediterranean by smaller structures (∼5 km radius) so-called Submesoscale Coherent Vortices (SCVs), likely formed along the western coast of Sardinia by the influence of bathymetry on the northward LIW flow. These observations have revealed the efficiency of these circulation features to transport warm and salty LIW from the boundary circulation toward the basin interior, and in particular across the Algerian Basin.

    Regarding the evolution of the hydrology, clear increases in temperature and salinity of all water masses have been observed

all over the Mediterranean sea and focusing on LIW, a warming of 0.005-0.007° C.year$^{-1}$ and a salinification of 0.0018 year$^{-1}$ have been observed between 1960's and 1990's in the Algero-Provençal Basin (Béthoux et al., 1990; Béthoux and Gentili, 1996, 1999). Schroeder et al. (2017) reported stronger trends of LIW temperature and salinity, respectively of 0.024° C.year$^{-1}$ and 0.006 year$^{-1}$ (1993-2016) using a mooring in the Sicily Channel. Margirier et al. (2020) have reported even larger increasing trends in the Ligurian Sea during the 2007-2017 period, respectively of 0.06±0.01° C.year$^{-1}$ and 0.012±0.02 year$^{-1}$, the

discrepancies likely due to the different periods and locations of such studies.

    In this study, we present new observations of the Algerian Gyres, highlighting their effect on the LIW distribution and propagation of thermohaline signals across the basin in a broader time period. We provide estimates of the trends over the last 60 years during three main periods and eight regional boxes. The timeseries are then used to infer circulation timescale and pattern

of LIW in the Algerian basin. We first present data and methods in Sect. 2 then, the results in Sect. 3, that we discuss in Sect. 4 and finally conclude briefly in Sect. 5.

## 2   Data and Methods

### 2.1   *In-situ* data

All available temperature and salinity profiles (from 1960 up to 2017 included) in the Mediterranean Sea coming from multiple platforms (Conductivity Temperature Depth (CTD) sensors, profiling floats, Gliders, Expendable Bathythermographs (XBTs) and Mechanical Bathythermographs (MBTs)) have been gathered from different sources in order to track the changes in the water mass properties, and make up-to-date climatologies of LIW in the Algéro-Provençal basin.

The hydrographical data used in this study were gathered from the Coriolis project (see $http://www.coriolis.eu.org/$), MEDAR/MEDATLAS (Fichaut et al., 2003), World Ocean Database (Conkright et al., 2002), MMD (Mediterranean Marine Data): collaboration between CNR (Consiglio Nazionale delle Ricerche) and ENEA (Italian National Agency for New Technologies, Energy and Sustainable Economic Development) (Borghini et al., 2019; Durante et al., 2019; Ribotti and Borghini, 2008), "EGO" (Everyone's Gliding Observatories $http://www.ego-network.org/$) and SOCIB websites ($http://www.socib.eu/$). A significant part of data is coming from MOOSE and SOMBA networks (Testor et al., 2010, 2017; Coppola et al., 2019; Mortier et al., 2014; Iudicone et al., 2014), that aim at addressing scientific and environmental issues relevant





for climate change in the northwestern Mediterranean Sea and in the Algerian Basin.

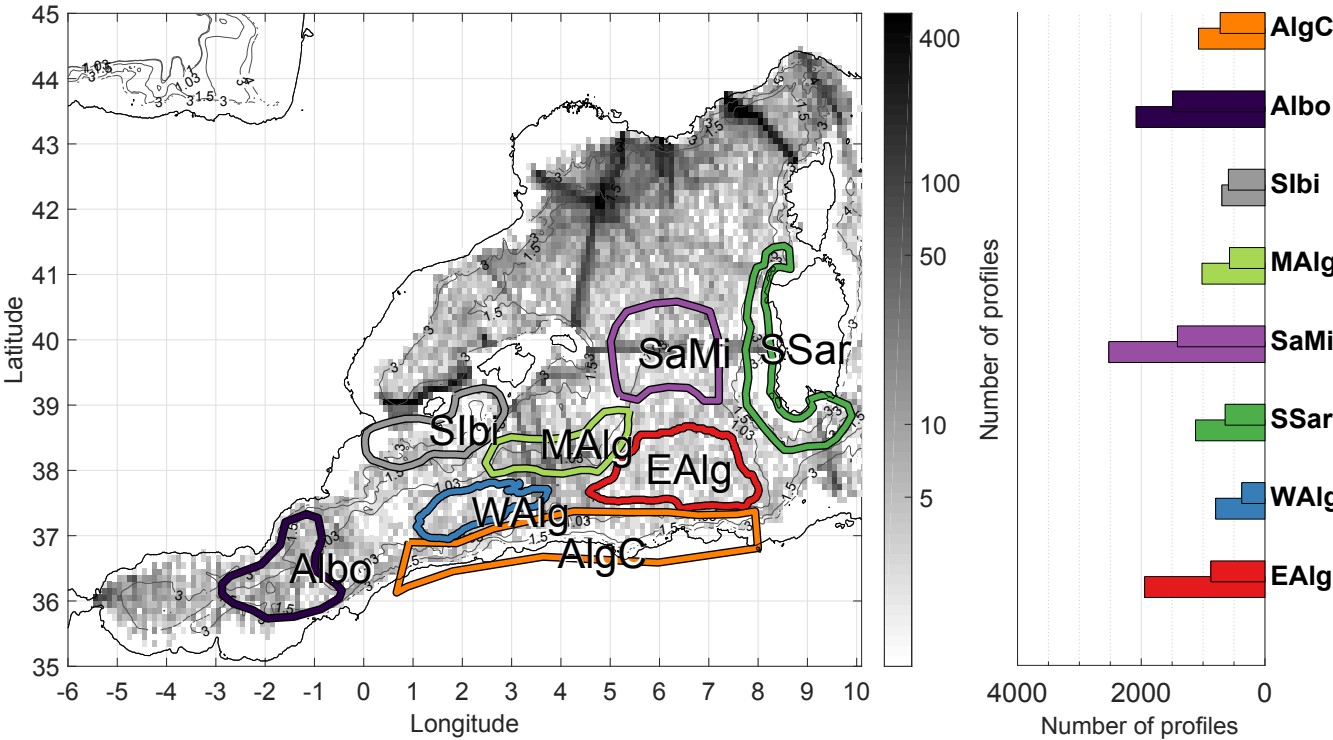

**Figure 1.** Maps showing number of LIW salinity data in the western Mediterranean Sea between 1960 and 2017 (left panel). The polygons define the areas used to select data for our comparative study. The bar plot on the right panel shows the total number of data in each polygon (bottom bar for $\Theta$, top one for salinity). The labels of the corresponding polygons are written in the y axis.

We applied rigorous and systematic quality controls and corrections on temperature and salinity data, in particular for XBTs,
as described by Houpert et al. (2015), allowing us to detect interannual variations at the basin scale with enough confidence.
Some additional quality controls, tailored for the Algerian basin, based on visual check have been applied:

**Correction of salinity offsets:** Temperature profiles are generally consistent in the deep layers (below ∼1000 m depth) where
the variability is small, and this provides good confidence in the upper measurements where the variability is higher.
However, salinity profiles can present a constant, sometimes large offset in the deep layers where variability is supposed

to be similarly small. We assumed they were due to sensor calibration issues and considered as profiles subjective to
offset correction the ones with a density larger than 29.13 or lower than 28.98 kg/m$^3$ between 800 and 1500 m (density
thresholds chosen outside ±4 standard deviations of natural variability in this layer of slow evolving characteristics over
depth). In order to correct them, the deep part of the salinity profiles were aligned with the mean regional deep salinity.
To this end, we used reference salinity data not concerned by the previous criterion, in the 1100-1500 m range within a





50 km radius and a 1 year time window. This 1100-1500 m layer was selected because it corresponds to WMDW having small natural variability within a year (0.01) compared to the corrections applied (typically 0.1-0.2) .This step ensures a relatively consistent data set in salinity.

**Removal of outliers:**  Based on climatological analysis previously published (Manca et al., 2004), and profile visualisations carried out with our data set, some profiles were considered as outliers and thus discarded if one of the following criteria
applies to them:

- Salinity larger than 39 or smaller than 36 below 100 m;

- Potential Temperature ($\theta$) larger than 17° C below 200 m, larger than 14° C below 1000 m, or smaller than 10°C;

- Potential densities larger than 29.2 kg/m$^3$ between 0 and 2000 m, or smaller than 28.5 kg/m$^3$ between 400 to 1000 m, or smaller than 29.02 kg/m$^3$ bellow 1000 m.

These quality controls and corrections result from many iterations and represent a trade-off between measurements accuracy and spatio-temporal coverage.

In addition to the potential temperature and salinity data, current measurements from SOMBA-GE 2014 research cruise (Mortier et al., 2014) were used. For this cruise, two 300kHz Acoustic Doppler Current Profilers (ADCPs) were attached to
the Rosette used to perform the CTD casts: namely LADCPs (Lowered Acoustic Doppler Current Profilers). The measured currents were processed using the velocity inversion method of Visbeck (2002) implemented in the LDEO software version IX-12 (Thurnherr, 2010) with typical horizontal velocity uncertainty of 2–3 cm.s$^{-1}$

## 2.2   Objective analysis of the LIW properties

To identify LIW, a density range between 28.95-29.115 kg.m$^{-3}$ was considered (red shaded area in Fig. 2). The temperature
and salinity maximum values were chosen to be representative of the LIW core characteristics for each profile.

To confirm that the water mass detected correspond to the LIW and not the base of the thermocline, we controlled the temperature maxima to make sure they actually correspond to an infexion point in the temperature profile.

One of the objectives of this study, is to describe a basin-scale mean repartition of LIW. To this end, we objectively analyzed the LIW salinity and temperature over the Algero-Provençal Basin. We first averaged the LIW salinity within 0.1° x 0.1° boxes,
and then analyzed this mean field using the method of Boehme and Send (2005) with a covariance function conditional to the topography and the planetary vorticity. We chose $\Phi$=0.5 as the scaling parameter representing the influence of the topography, and a spatial correlation scale of 100 km which is consistent with the basin-scale variability we want to emphasize similarly as Bosse et al. (2015).







**Figure 2.** Θ-S diagram of CTD casts performed within the polygons shown in Fig. 1 color-coded according to their dates. The black contours show isopycnals. The red shaded area (between 28.95-29.115 kg/m$^3$) is the zone considered to determine LIW characteristics.

### 2.3 Regions of interest

We chose 8 polygons (see Fig. 1) within the the Algéro-Provençal basin at key locations to characterize LIW along its pathway across the Algerian basin. The temperature and salinity profiles being in groups similarly typical of the different circulation features of the basin.

**Box EAlg:** closed f/H contour (f=8,9287.10$^{-5}$ s$^{-1}$, H=2797 m) in the Eastern part of the basin, typical to indicate the centre of the eastern Algerian Gyre away from the boundary circulation.





**Box WAlg:** closed f/H contour (f=8,9287.10⁻⁵ s⁻¹, H=2797 m) in the Western part of the basin, typical to indicate the centre of the eastern Algerian Gyre away from the boundary circulation.

**Box SSar:** polygon south of Sardinia. Inflow of warm/salty LIW into the Algéro-Provençal basin. This is where the warmest/saltiest LIW can be found in this basin.

**Box SaMi:** polygon between Sardinia and Menorca. Northern Edge of the eastern Algerian Gyre, where eddies detach from
the along-slope LIW vein.

**Box MAlg:** polygon in the Algerian basin centre at the northern periphery of the Algerian Gyres.

**Box SIbi:** polygon south Ibiza. Along-slope LIW circulation, at the almost end of its pathway toward the Gibraltar strait.

**Box Albo:** polygon in the Alboran Sea area. The LIW close to Gibraltar Strait about to exit the Mediterranean to form Mediterranean Outflow Waters in the north Atlantic or to recirculate along the continental slope of Algeria.

**Box AlgC:** polygon going along the Algerian coast. LIW entrained by Algerian Current that did not exit at Gibraltar Strait.

## 3 Results

### 3.1 Comparison of basin-scale CTD transects

Three East-West basin scale transects acquired during research cruises in the Algerian basin were available in our data set: MEDCO08 in November 2008 (Ribotti and Borghini, 2008), Venus1 in August 2010 (Borghini et al., 2019) and SOMBA-
GE2014 in August-September 2014 (Mortier et al., 2014). The comparable position and synoptic character of the cruise sampling allow for a direct comparison of this East-West section across the Algerian basin at these different dates over a period of 6 years (see Figure 3). Relatively warm and salty LIW extending far into the Algerian basin and away from the south Sardinian LIW vein can be observed. The signature of LIW fades away to the west as the distance from the source location, the Sardinia channel, increases, but one can identify a marked patch of LIW at about 400 km during each cruise (Fig. 3).
In addition to the information concerning water mass distribution across the basin, an increase in salinity (and temperature), illustrating the general salinification (and warming) trend of the basin, can be observed from one section to another. In the intermediate layer, the 38.52 isohaline appearing in the 2008 section is almost completely replaced by the 38.53 one in 2014, this is particulatly clear from 0 to 400 km from the benchmark A. Similarly, in the deeper layers, the isohaline 38.465 which surrounds a thick layer of less salty waters (between ∼1300 and ∼1800 m) in 2008, shrinks to a relatively small patch in the
eastern Algerian Basin in 2010 centred at 1500m depth, and then completely disappears in 2014. On the temperature panel, the isotherm 12.86° C evolves similarly.

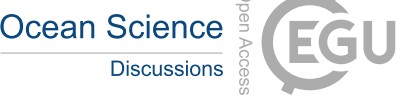

**Figure 3.** (a) Map of the locations of the CTD casts carried out during MEDCO08 in 2008 (green triangles, Ribotti and Borghini (2008)), during Venus1 in 2010 (orange crosses, Borghini et al. (2019)) and during SOMBA-GE2014 in 2014 (red dots, Mortier et al. (2014)). For comparison, all stations were perpendicularly projected on the A-B strait line. The faded colours are the actual locations of the casts, the bright ones represent their projections on A-B. East-West sections of (b,c,d) potential temperature and (e,f,g) salinity.





SOMBA-GE 2014 was the only research cruise specifically dedicated to investigate the oceanic circulation in the Algerian basin. To this end, a network of 70 hydrological casts have been carried out including direct measurements from surface to bottom of ocean currents using LADCPs (see Sect.2). Velocities averaged within different layers between 1200m and the bottom follow remarkably the f/H contours with a magnitude of about 5cm/s (blue arrows in Fig. 4).

Between 5° and 6° E, velocities are larger than 10 cm/s with a direction not matching the cyclonic circulation of the eastern Algerian Gyre (red ellipse in Fig. 4). This is due to the presence of a strong anticyclonic eddy at this location with a clear surface signature visible on SST and Ocean Color MODIS satellite images (see Fig. 5).

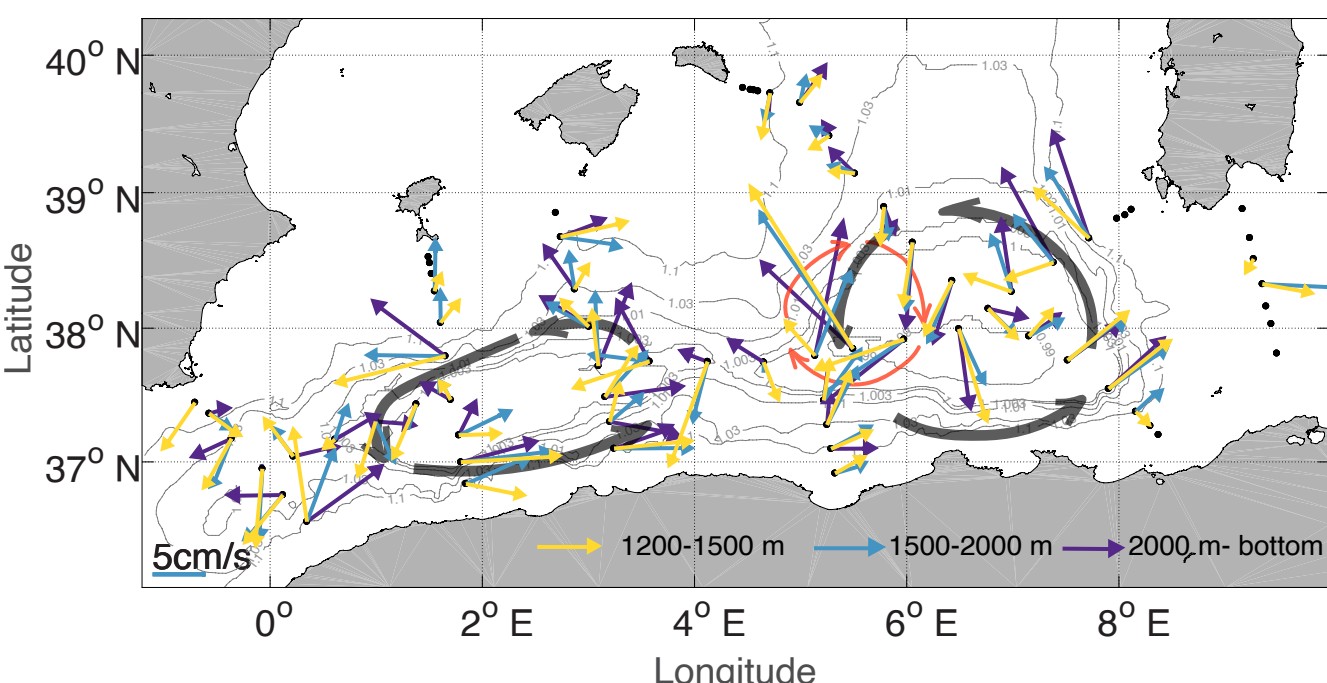

**Figure 4.** Map of all the CTD casts carried out during SOMBA-GE2014 cruise in 2014. LADCP measurements are indicated with arrows. They have been averaged within three layers: 1200-1500m (yellow), 1500-2000m (blue) and 2200m-bottom (purple). The black dots are the cast locations. The grey contours represent f/H contours, normalized by $f_0/H_0$ ($f_0$ being calculated at a latitude of 37°45' N, and $H_0$=2797m). The transparent black arrows represent the approximate position and the direction of the Algerian Gyres. The red arrows indicate the position of a strong barotropic anticyclonic Algerian Eddy during the campaign.

## 3.2 A climatological view from multi-platform *in situ* data

Figure 6 shows a map of LIW climatology in the whole western Mediterranean obtained by an objective analysis of the 106 780 potential temperature and 97 513 salinity LIW core values of the data set (1960-2017) and priorly averaged in 0.1 by 0.1° boxes.

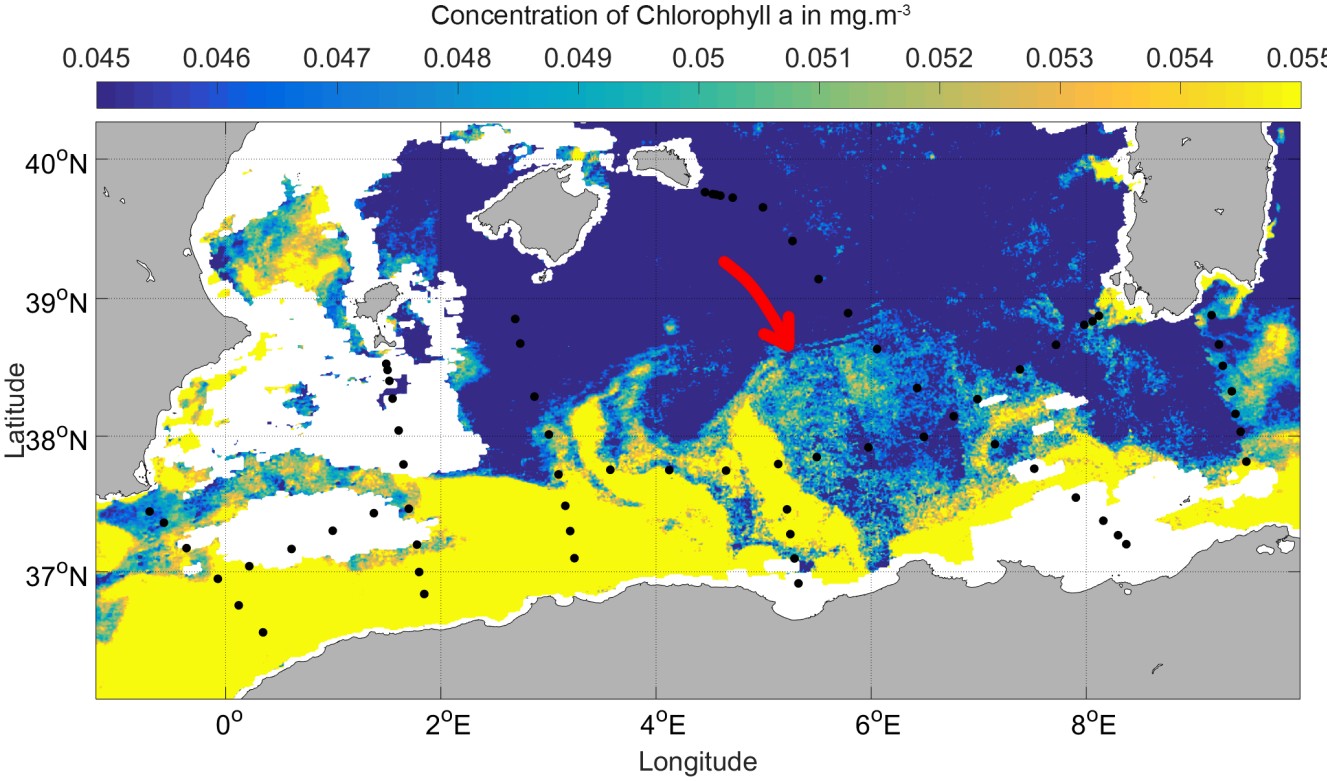

**Figure 5.** Level 2 Ocean Color image from MODIS, expressed in concentration of chlorophyll a (mg.m$^{-3}$), on the 25$^{th}$ August, 2014. The red arrow help identify the location of the eddy.

The warm and salty LIW vein can be observed along Sardinia and Corsica that further extends with the Northern Current in the Provençal basin. By looking at the 38.55 isohaline and the 13.3° C isotherm in Fig. 6, climatological warm and salty LIW can also be observed offshore and in particular north of the Western Algerian Gyre, extending from the LIW vein further offshore towards Menorca, then penetrating the interior of the Algerian basin roughly following the normalized f/H contour 1.03 that is represented by the white contour in Fig. 6.

Accordingly, the eastern Algerian basin is warmer and saltier than in the Provençal basin. Noteworthy, a warmer and saltier patch can be observed around 4° E, 38° N, it corresponds to ABACUS and AlgBasi glider missions that have been repeated from 2014 to 2018 and provided a lot of data inducing a bias in the average toward the recent warmer and saltier years.

In Fig. 7, a climatology of density in the western Mediterranean at 350 m (mean depth of the detected LIW in the western Mediterranean) shows the doming of the isopycnals in the north-western basin, with a maximum around the Gulf of Lion. This

reveals the cyclonic circulation of the Northern Gyre, characterized by a doming of isopycnals toward the surface allowing deep convection to occur (MEDOC, 1970; Testor et al., 2018). At the same depth, lighter waters are found in the Algerian



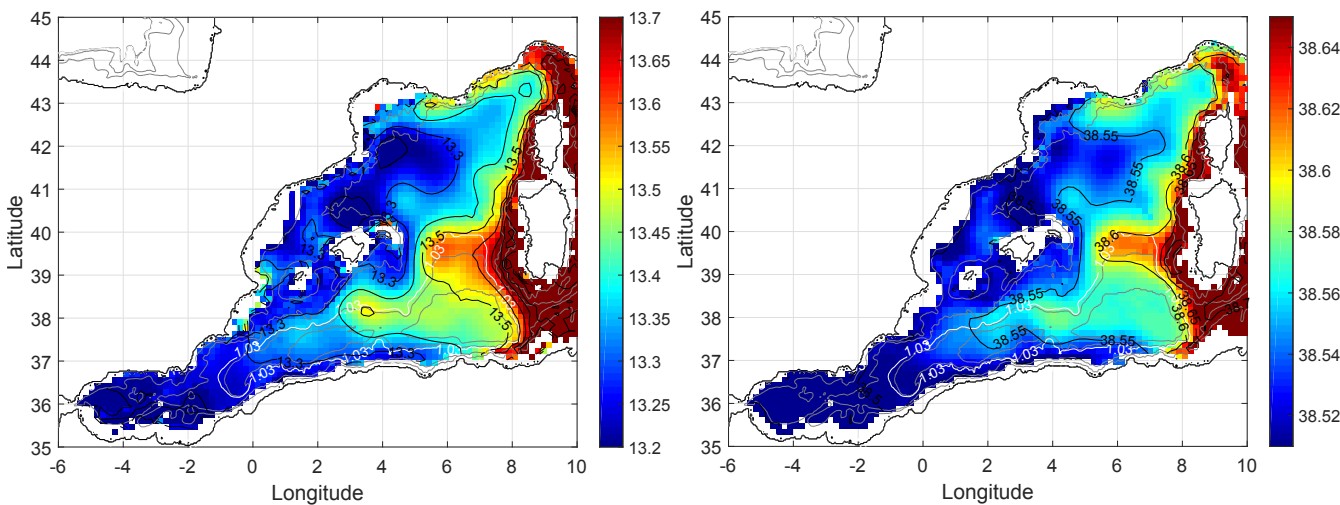

**Figure 6.** Climatologies of LIW potential temperature in °C (left) and salinity (right) obtained from an optimal interpolation of all the available data from 1960 to 2017, on a 0.1*0.1° grid.

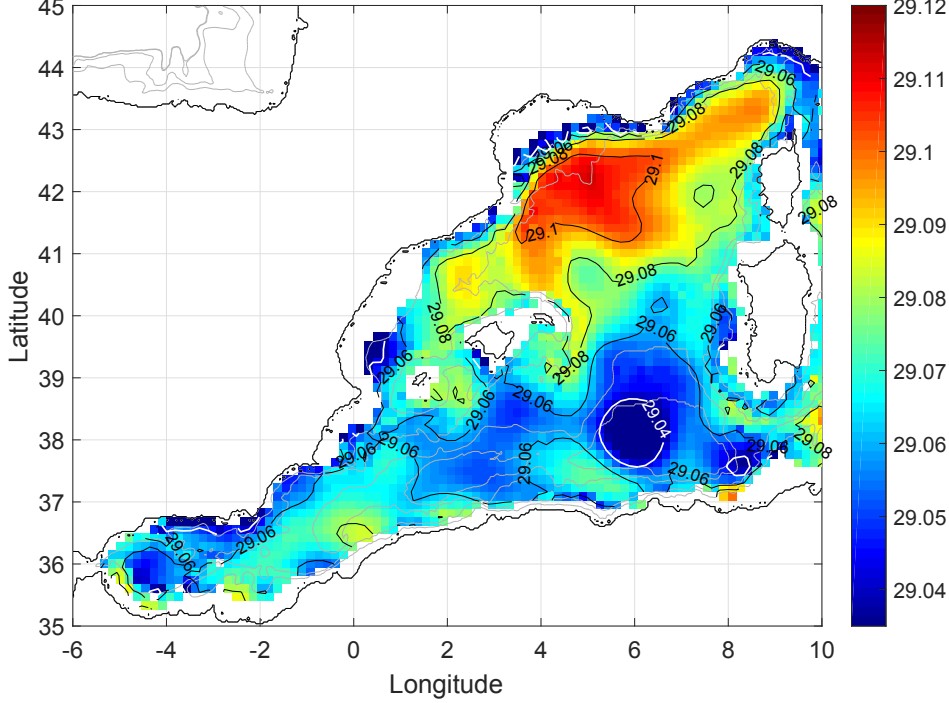

**Figure 7.** Same as Fig. 6 for water density at 350 m, in kg.m⁻³. The white contour 29.04 kg.m⁻³ is here to enhance the particularly remarkable sinking of isopycnals in the western Algerian Gyre.



Gyres, particularly in the centre of the eastern Algerian Gyre (see contour 29.04 kg.m$^{-3}$ in Fig. 7).

### 3.3 Temporal evolution of LIW characteristics

The evolution of potential temperature of the LIW as seen in Fig. 8 is showing an overall increase over the 1960-2017 period. However, this increase does not appear monotonous. The general shape of the timeseries suggests a roughly stable warming from the sixties to the eighties, followed by a decrease until the late eighties, then a significant increase after 2012. This would indicate 4 different phases in the basin regime (hereafter, period 1, 2, 3 and 4) during the full 1969-2017 period of our study. To estimate the trends in every phase, $\Theta$ is linearly fitted over time using least squares with a 95 % confidence interval, allowing
estimates of $d\Theta/dt$ and its uncertainty from the slope of the regression slope and its error. We also indicated the correlation coefficient $R^2$ for every regression.

In table 1, these potential temperature trends in each area were documented, during the different phases best fitted for each area.

To describe the salinity trends, the same method was used, the results are shown in table 2.
Here we will describe the evolution of the LIW temperature and salinity on average over the whole Algerian Basin for every period. (arithmetic mean of all the trends, with the standard deviation $\sigma$ associated).

– 1968-1978: During this $\sim$ 10 year period, an increase in temperature of about 0.011 $\pm$0.011° C.year$^{-1}$ and a salinity increase of 0.0022 $\pm$0.0058 year$^{-1}$ were observed.

– 1979-1987: A brutal decrease in temperature is observed in all the areas, with a strong regression coefficient( $r^2$>0.8). In four of the areas, the salinity data shows the same behaviour. This cooling signal is on average of -0.037 $\pm$0.007° C.year$^{-1}$ and the freshening signal is on average of -0.0024 $\pm$0.0081 year$^{-1}$.

One can identify the cooling event in Figure 8 as it appears first on the green curve (south Sardinia polygon) with an amplitude of about 0.3° C, then the signal propagates to the other areas.


– 1988-2012: During these $\sim$ 25 years, the temperature time series shows an irregular pattern, tending towards increase for most of the areas (6 regions over 8) with a mean of 0.0014 $\pm$ 0.0029° C.year$^{-1}$. Whereas the salinity time series has a clearer increasing trend of 0.0017 $\pm$0.0011 year$^{-1}$ on average with strong regression coefficients. We can also observe in some of the polygons, a drop in temperature right before the period of warming acceleration: SSar between 2005 and
2009, EAlg between 2008 and 2012 and WAlg between 2013 and 2014 after having plateaued (from 2010 to 2012).





**Figure 8.** (a) Mean potential temperature and (b) salinity of LIW core in the different areas in the southwestern Mediterranean Sea. The bar plots represent the mean number of data points in the polygons each year. the colour code used in this figure is the same as in Fig. 1. (standard error= standard deviation ( of $\Theta$ or S in one year)/$\sqrt{N}$, N being the number of $\Theta$ or S data within the year). The vertical blue-green dashed lines in background are indicators of the four periods chosen to compute the trends.



**Table 1.** Evolution of LIW temperature during different periods of time from 1960 to 2017, (expressed in mean trend $\pm$ the half width of the 95 % confidence interval °C.year$^{-1}$). The period slightly differs from one area to another, to best track the identified patterns. The coefficient of determination ($R^2$) is indicated, the regressions with an $R^2 < 0.5$ are indicated in gray.

| Box | full period | period 1 | period 2 | period 3 | period 4 |
|---|---|---|---|---|---|
| EAlg | 1967-2017 $R^2$=0.1 **0.0005** $\pm$0.0014 | 1967-1978 $R^2$=0 **0.0011** $\pm$0.0128 | 1979-1987 $R^2$=0.9 **-0.0266** $\pm$0.008 | 1988-2012 $R^2$=0.1 **0.0017** $\pm$0.0041 | 2013-2017 $R^2$=1 **0.0496** $\pm$0.0219 |
| WAlg | 1967-2017 $R^2$=0.2 **0.0008** $\pm$0.0013 | 1967-1979 $R^2$=0.8 **0.0122** $\pm$0.0057 | 1980-1987 $R^2$=1 **-0.0426** $\pm$0.0035 | 1988-2013 $R^2$=0.4 **0.0035** $\pm$0.003 | 2014-2017 $R^2$=1 **0.0862** $\pm$0.0276 |
| SSar | 1966-2017 $R^2$=0.1 **0.0006** $\pm$0.0021 | 1966-1977 $R^2$=0.3 **0.0051** $\pm$0.0079 | 1978-1986 $R^2$=0.8 **-0.0396** $\pm$0.0228 | 1887-2008 $R^2$=0.2 **-0.0015** $\pm$0.0022 | 2009-2017 $R^2$=0.9 **0.0573** $\pm$0.0229 |
| SaMi | 1967-2017 $R^2$=0.1 **0.0007** $\pm$0.0017 | 1967-1977 $R^2$=0.8 **0.0289** $\pm$0.0124 | 1978-1987 $R^2$=0.8 **-0.0388** $\pm$0.0186 | 1988-2012 $R^2$=0.3 **0.0022** $\pm$0.0027 | 2013-2017 $R^2$=1 **0.0712** $\pm$0.0276 |
| MAlg | 1968-2017 $R^2$=0.3 **0.0021** $\pm$0.0018 | 1968-1977 $R^2$=0.6 **0.0239** $\pm$0.0227 | 1978-1987 $R^2$=0.8 **-0.0312** $\pm$0.0184 | 1988-2012 $R^2$=0.1 **0.0007** $\pm$0.0041 | 2013-2017 $R^2$=0.8 **0.0436** $\pm$0.0399 |
| SIbi | 1969-2017 $R^2$=0.2 **0.0013** $\pm$0.0016 | 1969-1979 $R^2$=0 **-0.0011** $\pm$0.0179 | 1980-1988 $R^2$=0.9 **-0.0422** $\pm$0.0132 | 1989-2013 $R^2$=0.5 **-0.0034** $\pm$0.0021 | 2014-2017 $R^2$=0.9 **0.0889** $\pm$0.0763 |
| Albo | 1968-2017 $R^2$=0.6 **0.0033** $\pm$0.0012 | 1968-1979 $R^2$=0.5 **0.0065** $\pm$0.0067 | 1980-1988 $R^2$=0.9 **-0.0301** $\pm$0.0099 | 1989-2012 $R^2$=0.4 **0.0033** $\pm$0.0027 | 2013-2017 $R^2$=1 **0.0624** $\pm$0.0107 |
| AlgC | 1970-2017 $R^2$=0.6 **0.0044** $\pm$0.0015 | 1970-1982 $R^2$=0.7 **0.0146** $\pm$0.0081 | 1983-1988 $R^2$=1 **-0.0441** $\pm$0.0146 | 1989-2017 $R^2$=0.6 **0.0057** $\pm$0.0029 | —— —— |

– 2012-2017: Starting in 2012, the warming and salinification trends show a clear increase never reached before (see Fig. 8). The mean temperature trend in the Algerian basin during the full period is of $\sim$0.0017 $\pm$0.0014° C.year$^{-1}$, compared to a trend of 0.059 $\pm$0.017° C.year$^{-1}$ during these 5 years. In the same way, the salinity increased by one order of magnitude, from 0.0017 $\pm$0.0003 year$^{-1}$, to 0.013 $\pm$0.006 year$^{-1}$ during 2012-2017.

## 3.4 Transit time of LIW thermohaline signals

This section will be dedicated to quantify more thoroughly the transit time of the cooling signal observed in the 80s, using a cross correlation with a maximum lag considered of four years, of the signal between 1974 and 1992. In order to isolate this event on the time series, monthly averaged data, smoothed over four years were used.

In Fig. 9 the result of the cooling signal tracking across the Algerian basin, is presented. On the map, we can see in solid gray arrows, the along-slope circulation as shown in Millot and Taupier-Letage (2005b), the shear red polygons with the numbers, show in months, the time needed for the signal to travel from south Sardinia (SSar polygon) to the other areas in the Algerian basin.





**Table 2.** Same as table 1 for LIW salinity(unit)

| Box | full period | period 1 | period 2 | period 3 | period 4 |
|---|---|---|---|---|---|
| EAlg | 1967-2017 $R^2$=0.7 | 1967-1978 $R^2$=0.1 | 1979-1987 $R^2$=0.2 | 1988-2012 $R^2$=0.8 | 2013-2017 $R^2$=0.9 |
|  | **0.0012** ±0.0004 | **-0.0005** ±0.003 | **-0.0027** ±0.0104 | **0.0028** ±0.0008 | **0.0162** ±0.0091 |
| WAlg | 1967-2017 $R^2$=0.8 | 1967-1979 $R^2$=0 | 1980-1987 $R^2$=0.8 | 1988-2013 $R^2$=0.8 | 2014-2017 $R^2$=1 |
|  | **0.0016** ±0.0005 | **-0.0003** ±0.0067 | **-0.0100** ±0.0505 | **0.0023** ±0.0009 | **0.0150** ±0.0082 |
| SSar | 1966-2017 $R^2$=0.7 | 1966-1977 $R^2$=0.8 | 1978-1986 $R^2$=0.4 | 1887-2008 $R^2$=0.5 | 2009-2017 $R^2$=0.9 |
|  | **0.0019** ±0.0006 | **-0.0089** ±0.0072 | **0.0051** ±0.013 | **0.0014** ±0.0007 | **0.0136** ±0.0056 |
| SaMi | 1967-2017 $R^2$=0.8 | 1967-1977 $R^2$=0.8 | 1978-1987 $R^2$=0.9 | 1988-2012 $R^2$=0.7 | 2013-2017 $R^2$=0.9 |
|  | **0.0019** ±0.0005 | **0.0067** ±0.0279 | **-0.0158** ±0.0177 | **0.0020** ±0.0008 | **0.0217** ±0.0167 |
| MAlg | 1968-2017 $R^2$=0.7 | 1968-1977 $R^2$=1 | 1978-1987 $R^2$=1 | 1988-2012 $R^2$=0.5 | 2013-2017 $R^2$=0.9 |
|  | **0.0014** ±0.0006 | **0.0054** ±NaN | **0.0101** ±NaN | **0.0014** ±0.0013 | **0.0096** ±0.0059 |
| SIbi | 1969-2017 $R^2$=0.8 | 1969-1979 $R^2$=0.7 | 1980-1988 $R^2$=0 | 1989-2013 $R^2$=0.4 | 2014-2017 $R^2$=1 |
|  | **0.0020** ±0.0005 | **0.0063** ±0.0145 | **-0.0001** ±0.0059 | **-0.0007** ±0.0008 | **0.0174** ±0.0012 |
| Albo | 1968-2017 $R^2$=0.8 | 1968-1979 $R^2$=0 | 1980-1988 $R^2$=0.8 | 1989-2012 $R^2$=0.7 | 2013-2017 $R^2$=0.9 |
|  | **0.0015** ±0.0003 | **0.0002** ±0.0032 | **-0.0044** ±0.0029 | **0.0024** ±0.0012 | **0.0078** ±0.0065 |
| AlgC | 1970-2017 $R^2$=0.7 | 1970-1982 $R^2$=0.7 | 1983-1988 $R^2$=0.1 | 1989-2017 $R^2$=0.8 | —— |
|  | **0.0020** ±0.0005 | **0.0090** ±0.0052 | **-0.0010** ±0.0108 | **0.0024** ±0.0009 | —— |

In about two and a half years, the LIW travels from its source all the way to the Alboran Sea region. It appears that the fastest way goes from the Sardinia-Menorca polygon, SaMi (2 months) to the area between the Algerian Gyres, MAlg (4), then to the Alboran Sea ,Albo (29 months). Red arrows on the map represent a scheme of eddy-driven transport that could explain the transit times obtained from our analysis.

The signals arrive to the eastern Algerian Gyre, EAlg, after 23 months, to the western Algerian Gyre, WAlg, after 19 months and to the Algerian Current polygon, AlgC, after 37 months. The area that has the largest transit time is the one south of Ibiza (SIbi), 47 months.

The dashed circular gray arrows inside the Algerian Gyres represent the recirculation process in the core of the gyres.

Figure 10 illustrates the aforementioned cross correlation analysis. We can see how the cooling signal in the dashed curves (timeseries with lags) aligns remarkably with the cooling signal of the SSar timeseries.

In order to validate the results of the cross correlation analysis, a few pairs of time series segments have undergone the same analysis but for another time slot. Figure 11 shows the results obtained.

LIW potential temperature in South Sardinia have been cross correlated with the one in the Sardinia-Menorca region between

1990 and 2000, the result show that the signal needed 1 month to travel from SSar to SaMi, instead of 2 months in Fig. 9.



**Figure 9.** Circulation scheme of LIW in the Algerian Basin inferred from the cross correlation analysis. The transit time (indicated in months on top of each polygon) were obtained from the propagation of the cooling signal in the 1974-1992 period travelling from the South Sardinia area (green polygon). The colour of the polygons here is to provide a visual aspect of the time transit result, the redder the color, the larger the transit time. The solid gray arrows represent the mean along-slope intermediate circulation redrawn from Millot and Taupier-Letage (2005b). The solid red arrows represent the eddy-driven transport inferred from the result of the time transit analysis. The dashed gray arrows represent recirculation from the along-slope track.

Another cross correlation between MAlg and WAlg LIW potential temperature during the 1998-2010 period have shown that 13 months are needed for the signal to travel from MAlg to the interior of the western Algerian Gyre, instead of 15 months in Fig. 9. These results are in good agreement and suggest that similar processes are at play in the transit of the LIW independently of the period considered.





**Figure 10.** Cross correlation of the cooling signal (1974-1992) between SSar and all the other regions. The solid lines represent the original position of the timeseries. The dashed line represent the timeseries with a lag that give the maximum correlation. The lag, in months, is written above the curves.

## 4   Discussion

### 4.1   General circulation and LIW pathway

The results of the LADCP measurements presented in Sect. 3.1 show a current pattern that matches with the description of the Algerian Gyres done by Testor et al. (2005b) in terms of location and speed, however, the magnitude of the currents appear to be larger on the southern edge of both gyres (along the Algerian coast) and on the right edge of the eastern Algerian Gyre than on the remaining sides of the loops, suggesting that a forcing of these gyres is the general along boundary cyclonic circulation of the Western Mediterranean as discussed by Testor et al. (2005b). This result confirms the existence of the Algerian Gyres in



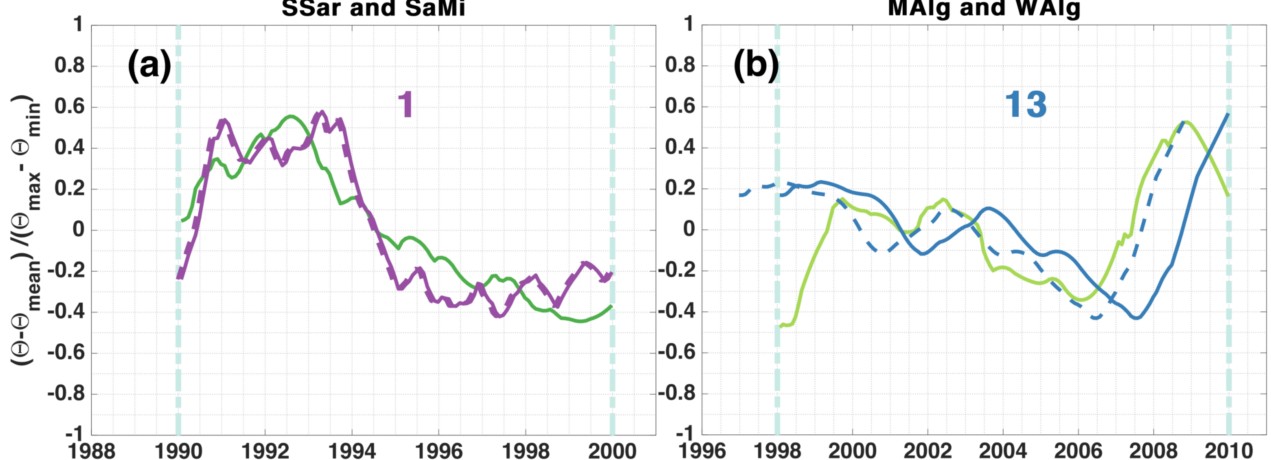

**Figure 11.** Cross correlation of normalized potential temperature signals. (a) SSar (green) and SaMi (purple) between 1990 and 2000, (b) MAlg (light green) and WAlg (blue) between 1998 and 2010. The solid lines represent the original position of the timeseries. The dashed line represent the timeseries with a lag that give the maximum correlation. The lag, in months, is written above the curves.

2014 as mean barotropic circulations that have signatures in the deep currents, and consolidates the idea of the Algerian Gyres being permanent circulation features.

The cruise sections from west to east in the Algerian Basin (Fig. 3) revealed changes in the hydrological distribution of LIW properties in the basin. Warm and salty LIW appeared to invade all the eastern Algerian Basin. The temperature and salinity climatologies of the LIW in the Western Mediterranean (Fig. 6) have also shown an influence of the Algerian Gyres on the LIW distribution. We can observe a good correspondence between the location of the 1.03 potential vorticity contour (a proxy of the Algerian Gyres) and the distribution of that warm and salty water extending further off-shore from the LIW vein. This
hydrological repartition have previously been observed, but was attributed to a slow accumulation over time of LIW in the interior of the Algerian basin, that remain unmixed, rather than a route of LIW crossing the Algerian basin (Millot, 1999). However, our study suggests that a direct route of LIW crossing the Algerian basin, linked to the presence of the Algerian Gyres, is instead likely to produce this effect.

From the climatological map of potential density at 350 m (Fig. 7), we can see a sinking of the isopycnals in the Algerian Gyres region. This may be the signature of numerous anticyclonic AEs, characterized by a deepening of isopycnals in their cores, circulating and accumulating in the basin.

In fact, in their study of coherent vortices in the Western Mediterranean using satellite altimetry, Escudier et al. (2016b, a) (1993-2012) and Isern-Fontanet et al. (2006) (1992-1999) observed intense anticyclonic eddies being particularly aggregated in



the Algerian Gyres area, and appear to follow the gyres' cyclonic circulation. This was also confirmed by Pessini et al. (2018), which used 1993 to 2016 altimetry data. Anticyclonic eddies were described by Puillat et al. (2002) to be the most energetic ones, capable of lasting several months to years, looping around the Algerian Gyres, some at least for 3 years. Provenzale (1999) evidenced that these vortices induce regular Lagrangian motion inside their cores and are highly impermeable to inward

and outward particle fluxes. Passive tracers can be trapped inside vortex cores for long times and are transported over large distances.

In the potential temperature time series (Fig. 8), one particularly strong cooling signal from 14.1 to 13.7° C could be identi-fied. It was then tracked across the basin as it progressed from east to west, using a cross correlation analysis.


The transit time analysis has shown a preferential path to get to the Alboran Sea region by entering the Algerian basin, from the northern edge of the eastern Gyre, then further flow south-westward at the periphery of the Algerian Gyres, as illustated by the thick red arrows on Fig. 9, and as seen in the climatologies on Fig. 6. The cooling signal was chosen to perform this analysis because it represents a particularly strong signal that appeared in all the timeseries, but the conclusions on the cir-

culation features are independent of this particular signal as they are governed by the internal dynamic of the basin. In fact, it corresponds to the eddy track that was observed in the multiple studies referred to hereabove (Testor et al., 2005b; Isern-Fontanet et al., 2006; Escudier et al., 2016a, b; Pessini et al., 2018). the anticyclonic eddies in the Algerian Basin cross from east to west with the Algerian Gyres's flow. There is also a resemblance with the Sardinia Eddies' track observed once by Testor and Gascard (2005) and modeled by Testor et al. (2005a), these eddies were observed to detach from the southwestern

corner of Sardinia, accumulate in the region here referred to as the Sardinia-Menorca polygon before being advected southward.

In the transit time analysis, the last area to get the signal was the south Balearic one, likely because this region receives most of its LIW from the along slope current of intermediate water and not much input from the most likely faster, eddy-driven cross-shelf transport. The intermediate water that gets to the south Balearic area is looping around the Northern Gyre first

before being advected south, it has also been affected by the convection occurring in the Gulf on Lion area, thus, the thermo-haline signals have been largely diluted. The transit time of 23 and 19 months obtained respectively in the eastern and western Algerian Gyres remain smaller than the transit time of the South Ibiza region but are relatively large considering their closeness to the LIW vein. This could be explained by the recirculation dynamics of the Algerian Gyres themselves added to the input from the AlgC region that alter the signal coming from the East.


## 4.2  LIW trends

The overall aspect of the temperature time series is very similar to the Western Mediterranean intermediate layer temperature evolution from Rixen et al. (2005) documented from 1950 to 2000. An increase from the 60s to 80s, followed by a drop lasting until the 90s, then a slower increase until 2000. The regression of the full temperature time series presents some positive trends,





however, the uncertainties are large (mean increase of $0.0017 \pm 0.0014°$ C.year$^{-1}$) and the correlation coefficient is small (R$^2$= 0.2 on average). Krahmann et al. (1998) and Rixen et al. (2005) reported the absence of a long term trend. However, positive trends in the intermediate water temperature from the sixties to the nineties have been shown by Béthoux et al. (1990) (0.005° C.year$^{-1}$) and Béthoux and Gentili (1996, 1999) (0.0068° C.year$^{-1}$). On the other hand, salinity trends for the full period are clearly toward an increase, $0.0017\pm0.0003$ year$^{-1}$. This result is similar to previous studies: increase of 0.0024 year$^{-1}$ during the

1955-1990 period (Rohling and Bryden, 1992) and 0.0018 year$^{-1}$ (Béthoux and Gentili, 1996, 1999) during the 1960-1992 and 1959-1996 periods, respectively. The temperature and salinity trends in the area between the Algerian Gyres (MAlg polygon) are very similar to the one reported by Vargaz-Yáñez et al. (2017) for the Balearic Sea sector between 1943 and 2015. These trends are of 0.002° C.year$^{-1}$ and 0.001 year$^{-1}$. They are however a little different from the results of our South Ibiza (SIbi) area (0.001° C.year$^{-1}$ and 0.002 year$^{-1}$).


The cooling signal observed during the late 70s, and start of the 80s in our study was reported by Brankart and Pinardi (2001). They showed that the origin of the phenomenon started in the Cretan Arc region, and have linked it to the heat flux anomaly evidenced by COADS time series. Krahmann et al. (1998) studied the temperature properties of the intermediate layer (275-475 m depth) during the 1955 to 1994 period, and a similar drop in temperature can be identified. This drop can

also be observed in the intermediate layer temperature timeseries in the studies by Vargaz-Yáñez et al. (2010a, b) and Rixen et al. (2005). In the latter paper, we see a similarity between the evolution of the timeseries of the Western Mediterranean at intermediate level, and the surface Eastern Mediterranean.

The third period (from 1988 to 2012) for which we have computed trends has an irregular pattern and the data coverage is

less regular than the other chosen periods. However, in some areas, we could identify a temperature drop after 2007. Zunino et al. (2012) have reported this event from the DYFAMED measurements in the Ligurian subbasin. They have linked this drop with the Western Mediterranean Transition, corresponding to changes resulting from the intense deep convection event that occurred in the Gulf of Lions and Ligurian subbasin in winter 2004-2005 (Schroeder et al., 2008, 2016).

The great acceleration of warming and salinification observed from 2012 to 2017, respectively $+0.059\pm0.017°$ C.year$^{-1}$

and $+0.013\pm0.006$ year$^{-1}$ have also been reported by Schroeder et al. (2017) in the Sicily channel between 2010 and 2016. They have recorded a potential temperature trend of $+0.064°$ C.year$^{-1}$ and a salinity trend of $+0.014$ year$^{-1}$. Barceló-Llull et al. (2019) documented similar trends in the Balearic sea between 2011 and 2018 ($+0.044\pm0.002°$ C.year$^{-1}$ and 0.010 year$^{-1}$). In Margirier et al. (2020) trends of $+0.06\pm0.01°$ C.year$^{-1}$ and $+0.012\pm0.02$ year$^{-1}$ between 2007 and 2017 were reported in the

Ligurian Sea.

Overall, the long term evolution of the temperature time series have allowed to identify a slow increasing trend from the sixties to 2017, but helped confirm the rapidly increasing trend after 2010.

## 5   Conclusion


Our study provides additional evidence that the Algerian Gyres represent an important circulation feature in the basin. It appeared on the current measurements that those gyres have an impact on the circulation over the whole water column. The study of the hydrological characteristics of LIW, using in situ data, showed that its distribution across the basin is linked to the presence of the gyres. A westward, cross-shelf, eddy-driven transport of LIW from the south Sardinia vein toward the interior
of the Algerian basin following the periphery of the Algerian Gyres is evidenced by the climatology of potential temperature and confirmed with the cross-correlation of a particular signal.

The LIW temperature and salinity trends estimates over various periods contribute to document LIW evolution in the Algerian basin and confirm the results of previous studies. More importantly, the warming acceleration that is observed all over the basin from 2010 is alarming. A closer monitoring of water mass properties need to be sustained, it is crucial to maintain and
reinforce existing surveillance systems as there is a direct impact on the regional climate and the marine resources.

*Author contributions.*   KM carried out the analyses, prepared the figures and wrote the main manuscript. HLG performed the processing of the current measurements. LH compiled the multisource in situ temperature and salinity data in one homogeneous product. KM and FM updated and improved the quality of the product. AB helped with optimal interpolation analysis. PT and AB provided guidance and supervision. LM and FL provided funding and administrative coordination. All authors have conributed in providing ideas, discussing the
results and reviewing the manuscript.

*Competing interests.*   The authors declare that they have no conflict of interest.

*Acknowledgements.*   We would like to particularly thank all crew members that have contributed to collect the precious in situ data. Thanks go to all scientists and technicians involved in the field campaign and data processing. We also thank Katrin Schroeder and her crew for providing the MEDCO08 and Venus1 CTD data.





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
