# Peer review of "The Levantine Intermediate Water in the western Mediterranean and its interactions with the Algerian Gyres: insights from 60 years of observation"

_Ocean Science, 2021_

## Author Response (AR1)

Answer to Comment on os-2021-120

**First, we would like to thank the Reviewers for their helpful comments. We think they have enabled us to improve the clarity of our manuscript. We are very grateful for these constructive feedbacks, especially concerning the statistical significance of estimated trends, and the overall readability of the manuscript. Please find below the individual responses to each Reviewer regarding their comments.**

**Referee Comments**
**Anonymous Referee #1**
Referee comment on "The Levantine Intermediate Water in the western Mediterranean and its interactions with the Algerian Gyres: insights from 60 years of observation" by Katia Mallil et al., Ocean Sci. Discuss., https://doi.org/10.5194/os-2021-120-RC1, 2022

Review of "The Levantine Intermediate Water in the western Mediterranean and its interactions with the Algerian Gyres: insights from 60 years of observation "
By Katia Mallil, P. Testor, A. Bosse, F. Margirier, L. Houpert, H. Le Goff, L. Mortier, and F. Louanchi.

The authors describe the Levantine Intermediate Water (LIW) in the western Mediterranean using in-situ data gathered over more than 50 years with a particular interest given to the Algerian Gyres region. The mean and variability of the LIW temperature and salinity are assessed. Regarding the variability of the LIW, the data suggests a significant cooling of the LIW in the late 70s early 80s and a rapid warming after 2012. Salinity trends are also described here (although in-situ are sparse). I found this work interesting, well-organized, and well-written. The amount of available in-situ data itself deserves publication.

While the analyses/trends would deserve more statistical significance (in particular with respect to the number of sin-situ data), I would recommend this article for publication after a careful revision of the following comments.

I wish the manuscript had included the origin and physical mechanisms of the cooling and warming. I hope it could be part of an additional study combining the large in-situ dataset presented here and some non-assimilated ocean model experiments.

Comments:
(1) Line 10: This does not seem to be a statistically significant trend for the period 1960-2017. In fact, you say warming for the period 1960-2017, but if you choose the period 1965-2012, it looks like a cooling trend. See below for more comments.

(2) **Thank you for bringing up this point, indeed, the obtained time series present a high variability and the value of the obtained trends depend highly on the chosen period, especially for temperature. In the previous version, we have used the annual means to compute the trends, verifying the statistical significance of the trends revealed that indeed, there was no significant temperature trend for the full period. and that was due to the use of annual means (low number of data points, and their high variability).**
(3) **In the revised version, we computed the regressions using the non averaged data, and although the data were highly variable, the long term trends remained significant. more details are given in the comments bellow.**

(1) Line 15: What does "active" mean in "active thermohaline circulation" ?

(2) **We qualified the thermohaline circulation as "active", because of the variety of processes involved, including deep convection. Also, the residency time of waters in the Mediterranean is estimated to be of about 100 years, which is an order of magnitude smaller than the residency time associated with global thermohaline circulation.**
(3) **We used instead the term "dynamic" also refering to recent changes in deep convection that were well documented (Margirier et al 2020, Somot et al 2018).**

(1) Line 24-25: Could you move these lines a couple of lines after? This paragraph is well written but you mention successively the MOW, then the LIW, then the MOW, and back to the LIW.
(2) **Thank you for pointing this out, it has been reorganised following recommendations of #Referee 2.**
(3) **After mentioning MOW, we proceeded to introduce LIW as an important constituent of it, describing its origin, properties and path. After that we indicated that the southwestern part of the Mediterranean was less documented than the other parts of the basin especially regarding LIW.**

(1) Line 29: "Levantine Intermediate Water " -> LIW
(2) **This have been sorted, thank you.**

(1) Line 37: If it does not make Figure 1 too busy and unreadable, could you show these Algerian gyres in Figure 1? Or perhaps having a new figure that shows the mean circulation of the region, including the Algerian gyres, Sardinia Eddies, bathymetry …. In particular for readers (like me) who are not familiar with this region.
(2) **Thank you for the suggestion.**
(3) **In the revised manuscript we have indicated the mean circulation of the LIW in the region in Figure 1, and represented a scatter of the data instead of a density gradient for the number of profiles to keep the figure readable.**

(1) Line 50: furtheR
(2) **Sorted, thank you.**

(1) Line 53: What are the findings of "Testor et al. (2005a)" ?
(2) **Testor et al. (2005a) have confirmed, using the numerical model, that the Sardinian eddies present a core of LIW at intermediate depths with characteristics close to those found in the Sardinian LIW demonstrating their transport efficiency.**
(3) **A sentence was added to clarify that.**

(1) Line 60: 0.005°C/year. Can it be called "warming"? I'm guessing that number is not statistically significant. Same for the salinity "trend". If the references listed here found these values to be significant, please add the information to the text.

(2) **Indeed the trends are small and based on few observations of a dynamical system. But still, these trends are however significant. The deep water properties were thought for a long time to have constant temperature and salinity, before oceanographers discovered that deep convection was renewing them and abruptly modify their TS properties. In that context, bethoux et al. (1990) have shown an increase of T of 0.12 °C between 1959 to 1989 in the deep water (>2000) using historical observations observation. Then from volume and heat conservation calculations, an increasing trend of 0.005 °C/year in the intermediate layer have been deduced. Bethoux et al. (1996) have compared in situ T and S measurements of intermediate layer from historical data covering the 1950-1973 period (Nyffeler et al. 1980) and measurements acquired in 1991 and 1992, and have shown an increase of temperature of 0.0068 °C/ year and an increase in salinity of 0.0018 / year.**

**Sparnocchia et al. (1994) have also reported a significant increase in LIW core temperature in almost all the areas of the Western Mediterranean, based on data from 1950 to 1987 (eg: 0.0091°C/ year in Ligurian Sea and 0.0065°C/year in Sicily channel).**
**These trends are synthesized in table 1 from Vargas et al. (2009)**
(3) **This explanation was added in the revised manuscript**

(1) Line 70-71: Sect. -> Section
(2) **In fact, it is in the OS instructions for authors that sections should be referred to by Sect.**

(1) Figure 1 caption: "total number of data" -> "total number of profiles"
(2) **Sorted, thank you.**

(1) L100: any references for the "WMDW having small natural variability within a year (0.01)". What is the percentage of data going through the correction of salinity offset?
(2) **After checking, it turns out that the correction of salinity offset algorithm failed to correct the detected outlying profiles in the absence of proper reference . We tried to adapt the parameters unsuccessfully then decided to discard this part. Thank you for pointing this out. The natural variability of ~(0.01) in salinity over a year was documented by Houpert et al (2016) during deep convection year, which can be considered as an upper bound, as years of weak convection will not affect the properties of the deep waters very much.**
(3) **The revised manuscript does not include a "correction of salinity offset" section.**

(1) L119: Reference for choosing the range "28.95-29.115 kg.m-3"?
(2) **We did not use literature to determine this range, instead, we used reference data such as quality-checked cruise and glider data to set a broad range encompassing the layer of LIW.**
(3) **A sentence was added to the manuscript to clarify that.**

(1) L136: Do you mean "western" Algerian Gyre?
(2) **Yes, thank you. The mistake have ben corrected.**

(1) L133-145: Except for EAIg and WAIg, how do you choose the other regions ? Are they related to bathymetry?
(2) **We chose the regions for our study to be relevant in respect to the circulation features.**
(3) **As suggested above and by reviewer 2, we have added on Figure 1 indications of the general circulation and the bathymetry that will help to understand our choices.**

(1) Line 154: Can you indicate the LIW in Figure 3?
(2) **Indeed the way sentence in line 154 (from the previous version) was not clear.**
(3) **The sentence have been rephrased to clarify what we meant, " but one can identify a marked patch of LIW at about 400 km during each cruise (Fig. 3) >> but one can identify a patch of higher temperature and salinity within the LIW layer, starting at about 400 km from point A, during each cruise (Fig. 3). A ellipse was added in Fig. 3 around the marked patch to help the reader.**

(1) Line 155-161: It would be clearer to plot the 2010-2008 and 2014-2008 differences (with a red-white-blue colorbar) to visualize the "trends".
(2) **Great suggestion, thank you**
(3) **A third column in figure 3 have been added to represent the differences $\theta_{2010} - \theta_{2008}$ and $\theta_{2014} - \theta_{2008}$ to visualise the evolution. Thank you.**

(1) Line 169: You might want to overlap SSH contours from altimetry to clearly see the anticyclone in Figure 5

(2) **Indeed, thank you for the suggestion.**

(3) **We have added , in Figure 5, contours of SSH from CMEMS product, the anticyclone is now clearly visible.**

(1) Line 172-174: Does the LIW exhibit seasonal variability? If so, how did you compute the climatological values?

**(2) LIW does not exhibit any seasonal variations, once formed in the Levantine basin, it is not in contact with the air, except in very specific areas of deep convection (NW Mediterranean Sea, Aegean, Adriatic, ...). In the study region, the LIW characteristics are mainly modified by mixing with the adjacent waters, and vertically by the interior turbulence.**

(1) Figure 6: That would be interesting (not for this manuscript, just a thought) to add to Figure 6 the thickness of the LIW.

(2) **In our study, we focused on detecting the core of the LIW, but for sure this would be an interesting perspective to look at the thickness of the LIW. Thank you for the idea.**

(1) Line 180-182: How do you compute the climatology. Did you just compute the mean of all your data? If so, and as you mentioned, the mean value will strongly depend on the number of data per year. Why not do the mean of monthly mean data? (Or if there is little climatology do the mean of yearly mean data).

(2) **In the previous version, the climatology was produced by computing a mean of all the data.**

(3) **We have proceeded to modify the method in the revised version, as suggested, we instead computed the mean of monthly means to reduce the bias, the patch of warm and salty LIW around 4° E that appeared on the previous climatology due to the large volume of glider data have disappeared with the change of method.**

(1) Line 188: Are the cyclonic Algerian Gyres some types of mode water eddies ?

(2) **No the Algerian Gyres are large scale cyclonic circulation features, mainly barotropic with a velocities of about 5cm/s within which the Anticyclonic Algerian Eddies live.**

(1) Line 190: from table 1, the "overall increase" is true only for Albo and AlgC

(2) **You are right, the trends of the whole study period were not very clear and significant in the previous version. In fact, the trends were calculated by doing a linear regression of the annual means that can be seen in Figure 8, the resulting trends were thus not very significant because they were based on few data points.**

(3) **In the revised version, trends were calculated using the totality of data, without any prior averaging, and P-values were computed to assess the statistical significance of the data. The results now show clearly the overall increase, even though the variability of the data result in low correlation coefficient values.**

(1) Line 193 and 202: Does your analysis start from 1969 or from 1960?

(2) **For the trend analysis, almost all the 60s data were discarded because of the scarcity of the data. However, in the climatology analysis, all data have been taken into account.**

(3) **A phrase have been added to explain that in line 208.**

(1) Line 194: Θ -> Θ & S
(2) **Sorted, thank you.**

(1) Line 195: By "error" you mean standard deviation?
(2) **We meant, confidence interval, the sentence is redundant, it has been rephrased. Thank you for pointing it out.**
(3) **To estimate the trends in every phase, θ is linearly fitted over time using least squares with a 95 % confidence interval, allowing estimates of dθ/dt and its uncertainty from the slope of the regression slope and its error >> To estimate the trends in every phase, the non-averaged θ is linearly fitted over time using least squares with a 95 % confidence interval, allowing estimates of dθ/dt and its uncertainty.**

(1) Table 1 and 2: Rather than using grey values for $R_2 < 0.5$, you should use grey values for non statistical significant values.
(2) **You are right, assessing the significance of the trends in the previous version by computing p-values revealed that none of the grey data ($R_2 < 0.5$) were statistically significant. In addition to that, some trends with $R^2 > 0.5$ were also not statistically significant, this was due to the low number of data points used for the regressions (we were using annual means), and their high variability.**
(3) **In the revised version of the manuscript, we considered the whole LIW data set, witout prior averaging, to compute the trends, and only considered as statistically significant those with a p-values inferior to the acceptable 0.05 threshold (the non significant ones being represented in gray in Table 1 and 2). Now, we obtained statistically significant increasing trends in 7 out of the 8 areas for the full period. Although the $R^2$ remain small, because the long term trends do not explain much of the variance of the data.**

(1) Line 197: "... different phases best fitted ...". I am not sure I understand. Why do you change the time period in each phase? For e.g. Why do you choose 2009-2017 for SSar? This is twice as long as the 2014-2017 period.
(2) **Indeed, the recent increase in temperature (already documented in the Sicily channel, see Shroeder et al 2017) arrived from the Easter Mediterranean entering the study area by the Ssar region. This is why we observe an accelerated increase in temperature already from 2009 in Ssar box. This is the motivation for having a different time period between the regions.**

(1) General comment: The number of points used in this analysis is very small, in particular for salinity for the period 1960-2010 or for temperature for the periods 1960-1970 and 2000-2010. In most years between 1960-2010 you have less than 5 data points per year for the 8 regions, meaning that most of the regions do not have data and the rest have 2-5 data points. This would be even more problematic if there is a seasonal cycle. How do you take this lack of in-situ data into consideration in your calculations?
(2) **First, we would like to clarify that the histogram that is represented in figure 8 represents the mean number of data used to compute the annual mean in each polygon, so there is on average 5 data points in each region. It is true that this number is not high, but over the whole period the number of data point become significant to make some statistics and compute some trends.**
**We have already justified why the seasonal cycle should not be an issue.**
(3) **In the revised manuscript, using non averaged data to assess the trends and computing p-values have adressed the issue of lack of data in certain areas. The trends in the areas with insufficient data were found not statistically significant (as can be seen for period 1 in table 1 and for period 1 and 2 in table 2) . We thus have a more realistic representation of the evolution of the water characteristics.**

(1) Line 202-203: Only half of the region shows that "increase". Also, Table 2 shows some misleading values. For e.g, the salinity increase for SaMi for the period 1967-1977 ($R_2$=0.8). It seems however that there is no data between 1970-1997 from Figure 8b.

(2) **You are right, in the previous version, there were misleading values regarding the trends.**

(3) **In relation to the previous comment, the change in methodology for the trends estimation and the computation of p-values have led us to avoid those mistakes. One can see now in table 2 that for the SaMi example discussed above, the salinity trend is no longer significant.**

(1) Line 207: Only 3 out of the 8 regions show a "significant" freshening. Yet, the conclusion is that there is a freshening.

(2) **Thank you for pointing this out, we have over stated our conclusions in this part.**

(3) **In the revised version we specified where a statistically significant freshening is observed, and as suggested by reviewer 2, we have computed a basin averaged trend estimate. For that we selected data inside a polygon roughly following the 2500m isobath to assess the trends in the Algerian basin interior. and it turns out that in the basin interior the freshening is also statistically.**

**The paragraph was modified as follows: A brutal decrease in temperature is observed in all the areas, with a strong regression coefficient( r2>0.8). In four of the areas, the salinity data shows the same behaviour. This cooling signal is on average of -0.037 ±0.007°C.year$^{-1}$ and the freshening signal is on average of -0.0024 ±0.0081 year$^{-1}$year-1>> A prominent decrease in potential temperature is observed in all the areas, with a relatively strong regression coefficient ($R^2 \geq 0.5$ in most of the areas). The salinity data also shows a decrease in three of the areas (WAlg, SaMi and Albo), but in SIbi the increase on salinity is not disrupted during this period. This cooling signal is on average of -0.033 ±0.003°C.year$^{-1}$ and the freshening signal is of -0.0037 ±0.0011 year$^{-1}$ in the basin interior.**

(1) Line 209: "propagates". How do you arrive at that conclusion? You look at yearly averaged, does it take several years for the signal to propagate from SSar to the other regions?

(2) **If we anticipate the results presented in the paper (fig 9), it takes indeed 1-3 years for the signal to propagate from the Ssar to the other region. If it was advected by horizontal currents, the signal would spread faster, but the cross-shelf exchange is mainly driven by horizontal diffusion by mesoscale eddies and thus takes more time than the typical advection speed of currents.**

(1) Line 211: If we assume that $R_2$<0.5 is not significant (I recommend using grey values for statistical significance), then only 2 out of the 8 regions have significant temperature trends (one warming and one colling). Yet, the conclusion here is "tending towards increase for most areas (6 regions over 8)"

(2) **Thanks for pointing this out.**

(3) **After changing the method for trend estimating (using non averaged data instead on monthly means), and assessing their significance by computing p-values, the trends of potential temperature during period 3 are indeed tending toward increase in most of the areas, the 6 over 8 regions is now a correct statement.**

(1) Line 215: "WAlg between 2013 and 2014 ". This is only a 2 year long period (i.e. 2

points).
(2) **You are right, this is not significant**
(3) **It has been removed from the text.**

(1) Figure 8: This figure is great and clean. The separation between bars (and points) which represent each year is however not consistent. It would be also easier to read if the vertical dotted lines also correspond to 1 year.
(2) **Indeed, thank you for the remark, it has also been pointed out by reviewer2.**
(3) **In the revised manuscript, we have applied both suggestions: vertical dotted lines now correspond to 1 year, and the spacing between bars and annual means is now consistent from one year to another.**

(1) Line 225: "monthly averaged data". I felt already that there were very few data points for the yearly averaged time series (bars on Figure 8a). This section now uses monthly averaged data. Do you have enough data for that?
(2) **The motivation for using monthly averaged data, is the possibility of obtaining smooth timeseries (after filtering) that faithfully display the large scale variations observed in the annual mean timeseries. It is true that the data are very scarce before 2010, but the interpolation and the filtering have allowed us to obtain timeseries usable for a cross correlation analysis.**

(1) Line 226-229: Need to be rephrased.
(2) **Thank you for pointing this out.**
(3) **This has been rephrased: "In Fig. 9, the cooling signal across the Algerian basin is tracked in time. The map shows in solid gray arrows the along-slope circulation, as shown in Millot and Taupier-Letage (2005b), the transparent red polygons with the numbers showing the time in months needed for the signal to travel from south Sardinia (SSar polygon) to the other areas in the Algerian basin."**

(1) Section 3.4: This is an interesting section. Are the "advection times" observed in other studies (could be during different periods of time). Can the 15months lag between MAIg and WAIg be explained physically? Is the surface circulation similar to the one at the LIW depth?
(2) **The circulation at LIW depth is clearly less energetic than at the surface, especially in the Algerian Basin where lots of intense Algerian Eddies have surface intensified velocities of ~(0.5m/s). The velocities in the LIW layer is ~(0.01-0.1m/s). The two regions are separated by approximately 200km, which corresponds to a propagation speed of 5 mm/s. There is no direct route to transport the LIW signal by the mean circulation and the eddy-induced horizontal transport act at an effective speed much smaller than the actual observed current velocities at the LIW depth.**

(1) Line 266: "have" -> "has".
(2) **Sorted, thank you.**

(1) Line 266: Is "Millot (1999)" support the "slow accumulation over time of LIW in the interior of the Algerian basin"? If so, move the reference up in the sentence.
(2) **Thank you for pointing this out , Millot (1999) does not explicitly talk about slow accumulation over time, it is only said that averaging without precautions may mislead into thinking there is a direct route from southwest of Sardinia towards the Algerian basin.**
(3) **Wording have been adjusted to specify author's intention.**

(1) Line 283: Add the period of time of the cooling discussed here.
(2) **Thank you for pointing this out.**

(3) **The period 1978-1986 have been added to the text, thank you.**

Line 292: "the" -> "The"
(2) **Sorted, thank you.**

(1) Line 292-295: What are the propagation speed of those eddies. Is it consistent with the lag correlations in Figure 9?.
(2) **The zonal velocities of large anticyclonic eddies detected and tracked from altimetry maps for the 20 year as estimated by Escudier et al. (2016) are of about 3 to 6 cm/s, (or ~2.5 to 5 km/day). if we use this number to estimate the propagation time of the signal, we obtain a transit time of ~1.5 to 3 months to cross 2° of latitude, which is consistent with the result we obtained for the SaMi region and the MAlg one.  Testor et al. (2005a) have estimated an average translation velocity of the Sardinian Eddies to be of ~ 2 to 3 cm/s (or ~ 1.7 to 2.5 km/day) which gives us, 3 to 4 months to cross 2° of latitude, this is slightly larger than the result we obtained for SaMi and MAlg, but the order of magnitude is consistent. The interior of the Algerian gyres however, present much larger transit times, and that is because of the wiggly motion that most of the eddies have following eddy-eddy interaction and the cyclonic barotropic circulation close to the gyres centers.**
(2) **This explanation have been added to the manuscript.**

(1) Line 296: Again, is the 47 lag realistic from the circulation of the region?
(2) **If we use the velocities of the LIW in the Provençal basin (5.7 to 9.4 cm/s) obtained from Margirier et al. (2020), to compute a transit time of a signal traveling from SSar to SIbi (~2450 km), we would obtain ~10 to 20 months, which is smaller than the 47 months obtained in our analysis, but the interesting information here is obtained when comparing the 47 months of SIbi with the 29 months in Albo that could reveal that the Eddy transport helps the efficiency of the intermediate water mass transport from SSar to Albo across the Algerian basin, and/ or the differential effect of this eddy transport. North of the eastern Algerian gyre, the transport is very effective, hence the 2 and 4 months, but then, once the eddies start swirling around the gyres, they progress slowly to transport the signal in other parts of the basin.**

(1) Line 354: Why is it alarming? Impact of biology, ecosystem, …?
(2) **Thank you for pointing this out.**
(3) **We have expended this last sentence: "A closer monitoring of water mass properties need to be sustained. It is crucial to maintain and reinforce existing surveillance systems as they can assess the direct impacts of climate change in the Mediterranean hot-spot. In the future, we can expect important modification of the water masses properties with major consequences: increase of temperature, stratification, collapse of deep convection in the NW Mediterranean Sea (Parras-Berrocal, et al 2022), thus affecting its profound functioning and the rich but fragile ecosystems that is hosts. It is reported in Lacoue-Labarthe et al. (2016) that an increased warming is likely to result in mass mortality of seagrass Posidonia oceanica (which is a very important habitat in the Mediterranean, and constitutes an important carbon sink), invertebrates, sponges and corrals ..etc. Invasive warm water species of algae, invertebrates and fish are increasing their geographical ranges. In addition to that, the proliferation of pathogens are expected, increasing the spreading of diseases."**

(1) *Acknowledgments*: Do not forget to add the funding agencies, if applicable.

(2) **We entered the information about funding separately in a dedicated space during submission. I suppose it will be included in the final version even though it is not visible in the pre-print**

**Anonymous Referee #2**

Referee comment on "The Levantine Intermediate Water in the western Mediterranean and its interactions with the Algerian Gyres: insights from 60 years of observation" by Katia Mallil et al., Ocean Sci. Discuss., https://doi.org/10.5194/os-2021-120-RC2, 2022

General comments

The ms presents an analysis of an extensive amount of thermohaline data in the Algerian basin from the 60s to almost the present day. From this large dataset, Mallil and coauthors investigate the role of the Algerian Gyres in the transport of the Levantine Intermediate Water (LIW) throughout the basin. Dividing the basin into distinct zones, they estimate temperature and salinity trends in the LIW density classes over different time periods. From a cooling signal detected in the LIW during one of these periods, first off Sardinia and subsequently in the rest of the considered regions, they estimate LIW spreading time periods throughout the Algerian basin.

The ms is rigorous and the conclusions are well supported. It is also well organized and clearly written. The wording of some sentences seems odd to me, but this rarely happens. I urge the authors to revise the use of symbols and units throughout the manuscript (see comments).

In my opinion, this paper is a significant and relevant contribution to the research community, and deserves to be published once the following comments and corrections are taken into account.

Specific comments

(1) L6. Indicate the values within the results section as well. It is stated on line 180 that the LIW is warmer and saltier in the eastern Algerian basin than in the Provençal basin, but the estimated values are not explicitly given there.
(2) **Thank you for pointing this out.**
(3) **The values have been added to the result section.**

(1) L6. Remove salinity units (practical salinity is unitless).
(2) **Thank you for pointing this out.**
(3) **All salinity units have been removed.**

(1) L6-L7. Is the sinking not presumedly produced by the regular presence of anticyclonic AEs coupled to the circulation of the Algerian Gyres (L270-L272)? I think it would be appropriate to clarify.
(2) **Thank you for pointing this out.**
(3) **This part have been rephrased for more clarity, thank you.**

(1) L14-L31. I recommend restructuring this part of the introduction slightly and checking the wording to improve clarity, especially for a reader unfamiliar with the topic and area of study. All the necessary information is already included there, but I find a bit confusing the way it is structured.

For example, one option might be: after explaining the thermohaline functioning of the basin, the exchange across the Strait of Gibraltar and the MOW, one could then describe the LIW as an important constituent of the MOW (describing the origin, properties and general circulation pattern of the LIW in the Western Mediterranean), to conclude by indicating that the circulation of the southwestern region of Mediterranean has been

relatively unexplored compared to other areas of the basin, especially regarding the LIW, even though the presence of the Algerian current and the generation of meanders and eddies are well known. Subsequently, in the next paragraph, proceed to describe the results of the MATER program in the region, and so on.

(2) **Indeed, as also pointed out by reviewer 1, this part needed reorganisation.**
(3) **It has been addressed following your suggestion.**

(1) L17. Relatively warm and fresh
(2) **This has been corrected, thank you.**

(1) L30-L31. LIW core is identified by an absolute salinity maximum and a relative temperature maximum.
(2) **It is indeed more precise with this formulation, thank you.**
(3) It **has been modified in the manuscript**

(1) L41. 'AEs transport LIW from the vein flowing northward along the continental slope of Sardinia…'. In relation to the previous comment (L14-L31), I think it would be helpful to include a general description of the LIW circulation in the Western Mediterranean so that the reader can more easily follow the introduction. Perhaps (if posible), even include a simple schematic in Figure 1 to help the reader? A subplot/inset?
(2) **Thank you for the suggestion, reviewer 1 also suggested that.**
(3) **In the revised manuscript we have indicated the mean circulation of the LIW in the region in Figure 1, and represented a scatter of the data instead of a density gradient for the number of profiles to keep the figure readable.**

(1) L64-65. Wording seems odd to me. Please, check it.
(2) **Thank you for pointing this out.**
(3) **This sentence have been rephrased for more clarity:the discrepancies likely due to the different periods and locations of such studies.>>The discrepancies of the trend values among those studies are likely due to the difference in the chosen periods and locations.**

(1) L66-L70. I suggest indicating also here briefly what type of observational data will be used in the study.
(2) **Thank you for pointing this out.**
(3) **A sentence have been added to this paragraph to mention the information, thank you.**

(1) Figure 1. Indicate in the caption the meaning of the background contour lines and specify units of the axes in the figure on the left. Also, use lowercase theta for potential temperature in the caption (correct this elsewhere in the manuscript and figures). Uppercase theta denotes conservative temperature.
(2) **Thank you for pointing this out.**
(3) **The corrections have been implemented, Thank you.**

(1) L103. General comment on 'Removal of outliers': why don't you set an upper density threshold for outliers in the deepest layers (>2000 m)? In Figure 2, one can observe a profile (from the 80-90s approx.?) that reaches a maximum sigma_theta value around 29.27 $kg/m_3$, and which I assume corresponds to the deepest layers. I understand that these values are outside your region of interest in the T-S plane, but why is this profile in Figure 2 not considered an outlier?
(2) **An upper density threshold have already been applied, any profile that presented density larger than 29.2 $kg/m^3$ from surface to 2000m, was considered an outlier. This particular profile that appears on figure 2 started presenting bizarre values precisely at 2000m, therefore, the criteria did not apply.**

(3) **The criteria was tweaked in order to address the problem. Thank you for mentioning that.**

(1) Linked to this comment, I think it would be convenient to indicate somewhere in the manuscript the maximum depth of the basin, just to contextualize the depth range of the profile.
(2) **Thank you for the suggestion.**
(3) **A mention to the maximum depth of the basin have been added in line 126.**

(1) L113. Since it has been indicated above (L107) that lowercase theta denotes potential temperature, you might use it here instead. At any rate, temperature and potential temperature are used interchangeably throughout the manuscript, therefore I recommend using one or the other consistently (if temperature is used, clearly state that it means potential temperature throughout the ms)
(2) **Thank you for mentioning that.**
(3) **It has been corrected throughout the manuscript.**

(1) L114. I suggest explaining here what these data will be used for.
(2) **The current measurements have been used to look for a signature of the Algerian Gyres reaching the deep layers.**
(3) **The information was added to the manuscript, thank you.**

(1) Figure 2. Remove salinity units (practical salinity is unitless), and replace uppercase theta (conservative temperature) with lowercase theta (potential temperature) as indicated above.
(2) **It has been sorted, thank you.**

(1) L129. General comment on 'Regions of interest': this subsection presents the distinct polygons defined from the different circulation features of the basin. The inclusion of a schematic in Figure 1 sketching those features as recommended before, would also greatly help the reader here.
(2) **Thank you for mentioning that.**
(3) **A scheme of the general circulation was added to Figure 1 to help the reader understand our choices.**

(1) L131-L132. This sentence sounds strange to me. Check it and consider rephrasing for clarity.
(2) **Thank you for the remark.**
(3) **The sentence have been restructured to clarify meaning (we chose the boxes to include temperature and salinity profiles being similar enough to characterize one particular stage of circulation of LIW).**

(1) L137. Replace 'south of Sardinia' with 'south and west of Sardinia' since the polygon extends all over that region, and not only off south Sardinia.
(2) **Thank you for mentioning that.**
(3) **It has been corrected throughout the manuscript.**

(1) L140. Replace 'along-slope LIW vein' with 'Sardinian along-slope LIW vein' for clarity.
(2) **Thank you for mentioning that.**
(3) **It has been replaced throughout the manuscript.**

(1) L148-L152. I suggest explaining this part at the end of Section 2.1.
(2) **Thank you for your suggestion.**
(3) **This part have been moved to the end of Section 2.1. as suggested.**

(1) L163-L165. I recommend including this information in the explanation of the SOMBA-GE 2012 survey provided in Section 2.1.

(2) **This part belonged indeed in Section 2.1.**
(3) **It has been moved there. Thank you.**

(1) L165/Figure 4. Why do you only use layers from 1200 m to bottom (well beneath the LIW core depth range)?
(2) **We chose to look at layers from 1200m down to the bottom, because it is more stable than the above layers with dynamical structures which velocities might conceal the lower velocity of the Algerian Gyres.**

(1) L177. Check 'Western Algerian Gyre'. Do you mean the Eastern Algerian Gyre?
(2) **Yes this was a mistake, thank you for pointing it out.**
(3) **It has been corrected.**

(1) Figure 6. Indicate also in the caption what the white isoline denotes. Add the axes units.
(2) **Thank you, this has been addressed.**

(1) Figure 7. Add axes units. Replace 'density' with potential density anomaly. Check 'Western Algerian Gyre' (Eastern?)
(2) **Again thank you for pointing out the mistakes**
(3) **They have been sorted.**

(1) L190. I wouldn't say 'overall increase' since that increase has a lot of uncertainty, only in 2/8 areas $R_2$ is greater than 0.5. The general increase is much clearer in the case of salinity.
(2) **Indeed, in the previous version we have used annual means of the LIW characteristics to compute the trends. Due to the high interannual variability of the tememprature compared to the long tem trend, no significant trend for the full period was detected.**
(3) **In the revised version, we computed the regressions using the non averaged data, and used p-values to assess their significance. Although the data were highly variable, the long term trends remained significant.**

(1) L197. Check wording.
(2) **Thank you for mentioning that.**
(3) **We added some clarifications to the sentence, thank you.**

(1) L205. 'A brutal decrease'. Wouldn't it be better to use another word such as 'prominent'?
(2) **In fact, prominent or noticeable are better suited in this sentence. Thank you for the suggestion.**
(3) **The word have been replaced.**

(1) L212. Indicate that in this period, potential temperature trends present very low $R_2$ values.
(2) **You are right, we should have referred to the trends in temperature with a comment on the low value of R2 for this period.**
(3) **In the revised version we have referred to the low regression coefficients and we have adjusted the interpetation of this part after assessing the significance.**

(1) L214-L215. This is a bit difficult to observe in the graph. I recommend changing the vertical grid in Figure 8 so that each dotted vertical line corresponds to the beginning of each year.
(2) **Indeed, thank you for the remark, it has also been pointed out by reviewer1.**

(3) **In the revised manuscript, we have applied both suggestions: vertical dotted lines now correspond to 1 year, and the spacing between bars and annual means is now consistent from one year to another.**

(1) Figure 8. Since monthly means are used in the cross-correlation, I understand that the vertical bars refer to the average number of points in EACH polygon (not in ALL polygons). The right y axis label in Figure 8 is a bit confusing to me. I may be missing something. Could you clarify this point?
(2) **In figure 8, the vertical bars represent the mean number of points used to compute the annual mean each year, independently of which polygon it is. We tried representing the number of data in each polygon, respecting the color code (8 bars every year), but it crowded the figure too much so we computed the mean value and represented it in grey. The bar plot is here to illustrate the increase of the observing capacities in the last 20 years or so, mainly due to the start of the Argo program.**

(1) Caption in Figure 8. Indicate that these are annual means.
(2) **The legend indicates that these are annual means.**
(3) **The information have been added to the caption as well. Thank you.**

(1) Table 1. Could you include the basin-averaged trend as well? Same for Table 2.
(2) **This is a very good suggestion, thank you.**
(3) **In the revised manuscript, we have computed a basin averaged trend estimate. For that we selected data inside a polygon (dotted dark gray contour in Figure 1) roughly following the 2500m isobath, to assess the trends in the Algerian basin interior, and it turns out that, as predicted, the increase of points in the data set have improved the significance of the trend.**

(1) Table 2. Check MAlg (NaNs in Period 1 and 2).
(2) **Thank you for pointing this out.**
(3) **We have fixed the bug.**

(1) L266. I think the reference should be placed earlier in the sentence.
(2) **Thank you for pointing this out.**
(3) **Wording have been adjusted to specify author's intention.**

(1) L270-L272. Just a comment: wasn't the presence of one of them captured during the SOMBA-GE 2014 survey?
(2) **Indeed an anticyclonic Algerian Eddy was captured during the SOMBA-GE2014 cruise, the surface signature of this AE is represented in fig.5, and the signature of its high velocities, even in the deeper layers, can be seen in fig.4. The East west section of the SOMBA-GE 2014 also shows a deepening of isotherms and isohalines at 600 km from point A on the panels (d) and (g) in fig.3.**
(3) **We added a reference to this observation in the text.**

(1) L299. Clarify that the cross-shelf transport occurs over the Sardinian continental shelf.
(2) **Indeed, we also noticed that this sentence was not particularly clear so we rephrased: "In the transit time analysis, the last area to get the signal was the south Balearic one, likely because in this region the LIW comes mainly from the along-slope advection by currents at intermediate depth circling the whole**

**Western Mediterranean Sea, and is not much influenced by the less efficient eddy-driven transport across the Sardinian shelf."**

(1) L309. Why 'some positive trends'? According to Table 1, all potential temperature trends are positive for the full period.
(2) **Indeed, thanks for pointing this out.**
(3) **After assessing the significance of the trends using P-values, we have specified in the table which data were statistically significant (significant trends in black, non significant trends in gray), the interpretations were modified accordingly in the text as well.**

(1) L.354. Explain why is it alarming.
(2) **Thank you for pointing this out.**
(3) **We have expended this last sentence: "A closer monitoring of water mass properties need to be sustained. It is crucial to maintain and reinforce existing surveillance systems as they can assess the direct impacts of climate change in the Mediterranean hot-spot. In the future, we can expect important modification of the water masses properties with major consequences: increase of temperature, stratification, collapse of deep convection in the NW Mediterranean Sea (Parras-Berrocal, et al 2022), thus affecting its profound functioning and the rich but fragile ecosystems that is hosts. It is reported in Lacoue-Labarthe et al. (2016) that an increased warming is likely to result in mass mortality of seagrass Posidonia oceanica (which is a very important habitat in the Mediterranean, and constitutes an important carbon sink), invertebrates, sponges and corrals ..etc. Invasive warm water species of algae, invertebrates and fish are increasing their geographical ranges. In addition to that, the proliferation of pathogens are expected, increasing the spreading of diseases."**

Technical corrections

(1) L8. Replace 'the signal' with 'a signal'
(2) **Sorted, thank you.**

(1) L15. Change 'is' to 'are'
(2) **Sorted, thank you.**

(1) L16. Preconditioning
(2) **Sorted, thank you.**

(1) L20. AEs acronym should be defined here, right after 'Algerian Eddies'
(2) **Sorted, thank you.**

(1) L23. water masses
(2) **Sorted, thank you.**

(1) L23. of the whole Mediterranean
(2) **Sorted, thank you.**

(1) L24. MOW acronym should be defined here, following 'Mediterranean Outflow Water'
(2) **Sorted, thank you.**

(1) L35-L36. Delete brackets in the coordinates
(2) **Sorted, thank you.**

(1) L41. Use AEs instead of 'Algerian Eddies' (also in L47)
(2) **Sorted, thank you.**

(1) L54. missing 'r' in further
(2) **Sorted, thank you.**

(1) L60. Sea
(2) **Sorted, thank you.**

(1) L68. broad
(2) **Sorted, thank you.**

(1) L109. Below
(2) **Sorted, thank you.**

(1) L119. Figure 1 must be referenced before Figure 2 (Figure 1 is referenced for the first time on L130. Section 2.3).
(2) **Thank you for pointing this out.**
(3) **A reference to Figure 1 have been added to the first paragraph of Section 2.1, thank you.**

(1) L120. Indicate that the 'maximum values' are maximum values within the selected range
(2) **Sorted, thank you.**

(1) L144. Use MOW instead of Mediterranean Outflow Waters
(2) **Sorted, thank you.**

(1) Figure 3. Add axes units in (a) and colorbar units in (b), (c), (d) or indicate them in the caption.
(2) **Sorted, thank you.**

(1) Caption of Figure 5: helpS TO identify
(2) **Sorted, thank you.**

(1) L193. Replace 1969 with 1960.
(2) **Thank you for pointing this out. In fact, we discarded the early 60's period for the trend estimation study, because of the scarcity of the data.**
(3) **In the revised manuscript, this was specified in the text**

(1) Caption of Table 1. Replace 'temperature' with 'potential temperature'
(2) **Sorted, thank you.**

(1) L227. Replace 'along-slope circulation' with 'basin-scale along-slope circulation'.
(2) **Sorted, thank you.**

(1) L227. What do you mean by 'shear red'?
(2) **I intended to type sheer red, meaning partly transparent, but it turns out that sheer is a specific adjective for fabric.**
(3) **In the revised manuscript the word transparent was used instead, thank you.**

(1) Table 2. Delete '(unit)' in caption.
(2) **Sorted, thank you.**

(1) Figure 9. Correct 'PRincipal'. Add axes units.
(2) **Sorted, thank you.**

(1) Caption of Figure 9. L2. Replace 'were' with 'was'

(2) **Sorted, thank you.**

(1) L254. Replace 'right' with 'easternmost'
(2) **Sorted, thank you.**

(1) L264. Indicate that the LIW vein is that off Sardinia.
(2) **Sorted, thank you.**

(1) L283-L284. Specify after 'could be identified' that it was south/west of Sardinia.
(2) **Sorted, thank you.**

(1) L292. The anticyclonic
(2) **Sorted, thank you.**

(1) L317. Vargas-Yáñez (check the spelling). Also in L325.
(2) **Sorted, thank you.**

(1) L342. has
(2) **Sorted, thank you.**

(1) L343. replace 'but' with 'and'?
(2) **Sorted, thank you.**

(1) L359. conTributed
(2) **Sorted, thank you.**